# Dermatological changes in a prospective cohort of acutely ill, hospitalised Malawian children, stratified according to nutritional status

Deborah van den Brink [1], Kelvin Mponda,[2] Debbie Thompson [3] Colette van Hees,[4] Fletchter Ngong'a,[5] Emma Segula,[5] Emmie Mbale,[5] Michael Boele van Hensbroek,[1] Robert H J Bandsma,[6,7] Judd L Walson [7,8] Daniella Brals,[9] James Berkely,[7,10,11] Wieger Voskuijl[1,5,7]

**To cite:** van den Brink D, Mponda K, Thompson D, et al. Dermatological changes in a prospective cohort of acutely ill, hospitalised Malawian children, stratified according to nutritional status. *BMJ Paediatrics Open* 2024;**8**:e002289. doi:10.1136/ bmjpo-2023-002289

For numbered affiliations see end of article.

**Correspondence to**
Dr Deborah van den Brink; d. vandenbrink@alumni.ubc.ca

## ABSTRACT

**Rationale** Since the first documentation of skin changes in malnutrition in the early 18th century, various hair and skin changes have been reported in severely malnourished children globally. We aimed to describe the frequency and types of skin conditions in children admitted with acute illness to Queen Elizabeth Central Hospital, Blantyre, Malawi across a spectrum of nutritional status and validate an existing skin assessment tool.

**Methods** Children between 1 week and 23 months of age with acute illness were enrolled and stratified by anthropometry. Standardised photographs were taken, and three dermatologists assessed skin changes and scored each child according to the SCORDoK tool.

**Results** Among 103 children, median age of 12 months, 31 (30%) had severe wasting, 11 (11%) kwashiorkor (nutritional oedema), 20 (19%) had moderate wasting, 41 (40%) had no nutritional wasting and 18 (17%) a positive HIV antibody test. Six (5.8%) of the included patients died. 51 (50%) of children presented with at least one skin change. Pigmentary changes were the most common, observed in 35 (34%), with hair loss and bullae, erosions and desquamation the second most prevalent skin condition. Common diagnoses were congenital dermal melanocytosis, diaper dermatitis, eczema and postinflammatory hyperpigmentation. Severe skin changes like flaky paint dermatosis were rarely identified. Inter-rater variability calculations showed only fair agreement (overall Fleiss' kappa 0.25) while intrarater variability had a fair-moderate agreement (Cohen's kappa score of 0.47–0.58).

**Discussion** Skin changes in hospitalised children with an acute illness and stratified according to nutritional status were not as prevalent as historically reported. Dermatological assessment by means of the SKORDoK tool using photographs is less reliable than expected.

## INTRODUCTION

Dermatoses are an important cause of morbidity in children in sub-Saharan Africa, with children under 5 predominantly at risk.[1] Most common skin conditions seen in rural or outpatient settings often have an infectious origin such as impetigo, tinea capitis or scabies.[2–4] Eczema is also commonly reported.[2 3] Children with oedematous malnutrition have been reported to have distinct skin and hair changes that differ from common dermatoses seen in the general population.[5–7] Nearly 90 years ago, Dr. Cicely Williams observed children who would die within a month of the skin changes.[5]

In severe wasting (SW), skin lesions, including angular cheilitis and pale mucous membranes, have been linked to micronutrient deficiencies, such as B vitamins and zinc.[8 9] Nutritional oedema ('kwashiorkor') has been linked to hyperpigmented skin patches in areas frequently exposed to pressure (groin, buttocks, knees and elbows).[6 7 10 11] Over time, the skin darkens, with a shiny, varnished appearance which

may desquamate leaving hypopigmentation, commonly referred to as 'flaky paint dermatosis'.[6 10] Hair may appear sparse, dry and brittle.[6 7 10 12] Intermittent malnutrition is characterised by bands of light and dark colouration in the hair termed 'flag sign'.[6 8 10]

**WHAT IS ALREADY KNOWN ON THIS TOPIC**
⇒ There is a lack of current studies reporting on skin changes in acutely ill hospitalised children in lower-middle-income countries.

**WHAT THIS STUDY ADDS**
⇒ Skin changes that were observed were common dermatoses not specifically linked to nutritional status.
⇒ The prevalence of skin changes in the subgroup of malnourished children was unexpectedly lower.
⇒ Historically reported dermatoses and hair changes associated with malnutrition were not as prevalent.

**HOW THIS STUDY MIGHT AFFECT RESEARCH, PRACTICE OR POLICY**
⇒ Skin assessment using clinical photography is challenging, subjective and may be less reliable than in-person assessments.
⇒ Future skin assessment tools in children in developing nations should also include common dermatoses.

Heilskov *et al* identified a lack of standardised terminology used in the literature making it difficult to describe the global prevalence or even correctly characterise the skin changes due to malnutrition.[13] They devised a scoring tool (SCORDoK, see online supplemental appendix 1) to identify skin manifestations and used it to predict mortality among children admitted to hospitals with severe malnutrition in Uganda.[14]

The primary objective of this study was to assess the prevalence of specific skin conditions among children across a spectrum of nutritional status admitted with an acute illness to the Paediatric Wards of Queen Elizabeth Central Hospital, Blantyre, Malawi (QECH). Our secondary objective was to validate the SCORDoK grading tool in a population of acutely ill, young children in Malawi.

## METHODS

Children aged 2–23 months old admitted to hospital and enrolled in the Childhood Acute Illness and Nutrition (CHAIN) Network cohort at QECH were eligible for the skin study.[15] CHAIN was a multisite, prospective, observational cohort study of children aged 1 week to 23 months old admitted to hospitals with an acute illness (November 2016–January 2019). The main aim of CHAIN was to identify causes and mechanisms of mortality in acutely ill children admitted to the hospital and for 6 months postdischarge. The CHAIN Network cohort study is described in detail elsewhere.[15 16] Briefly, participants were children, aged 2–23 months, with acute illness admitted to nine hospitals in six countries across sub-Saharan Africa and South Asia between 20 November 2016 and 31 January 2019. QECH (Blantyre, Malawi) was one of the nine CHAIN Network sites. In the entire CHAIN Network, 3001 children were enrolled in a 2:1:2 ratio in three strata (no wasting, moderate wasting (MW) and SW or nutritional oedema). CHAIN purposely over-recruited children with malnutrition, who are at higher risk of mortality.[17–19]

In the skin study, SW was defined using WHO criteria[20] as having a weight for length Z-score (WLZ) <−3 or an MUAC (mid-upper arm circumference) of <11.5 cm (6–59 months), MUAC<11.0 cm for infants aged <6 months. MW was defined as having a WLZ between −2 and −3 or an MUAC<12.5 cm, or MUAC<12.0 cm for infants aged <6 months. Non-wasting (NW) was defined as anthropometry above these thresholds. Nutritional oedema (kwashiorkor (KW)) was defined as the presence of bilateral oedema.

### Enrolment to main chain cohort

Due to extensive questionnaires and sampling, a maximum of five children per week, admitted to hospitals with an acute illness, were enrolled in three strata defined by anthropometry at each of the nine sites.[15] The CHAIN cohort dataset included a skin assessment for all enrolled participants by the admitting clinician.

### Enrolment to the skin study

Children who were enrolled on the main CHAIN cohort at QECH in Blantyre, Malawi between January 2018 and January 2019 were potentially eligible for the skin study. Enrolment was based on convenience sampling, dependent on the presence of the dermatologist on site, therefore, children who died or were discharged within 48 hours were sometimes missed. Children who were unable to leave their beds or on oxygen therapy were excluded due to the photography requirements. Specific informed consent for photographs was sought for the skin study as well as additional questions to establish pre-existing skin conditions (online supplemental appendix 2).

### Skin assessments

Children included in the skin study had photographs systematically taken of their entire body by an experienced medical photographer according to standardised dermatological documentation practices (online supplemental appendix 3). Photographs were taken on enrolment to the skin study and repeated 180 days postdischarge.

Children were clinically assessed by a trained and experienced dermatologist (KM) and skin conditions were documented. Pictures were stored on an encrypted external drive and transferred to dermatologists using 'FOX IT portal' providing multiple layers of encryption ensuring confidentiality.[21]

Training of SCORDoK tool was done prior to grading to ensure consistency in grading among dermatologists (table 1). Three dermatologists (KM, CvH and DT) with decades of experience with dark-skinned children and treating diseases in the tropics, scored each child's photographs according to SCORDoK tool definitions.[14]

Telogen effluvium was interpreted as a loss of hair and grading for pigmentary changes included any pigmentary change seen. Ichthyosiform skin changes referred to skin dryness while lichenoid skin changes referred to the thickness of skin. Bullae, erosions and desquamations referred to the blistering, denudation and peeling of the skin. Differences between observer scores were assessed by an independent author (DB) and cases with discrepancies in the scores were then discussed among the dermatologists resulting in a single agreed on score. Dermatologists reassessed photographs 6 months later to calculate intraobserver variability.

### Statistical analysis

Baseline characteristics were compared with $\chi^2$ and one-way analysis of variance using Python V.3.8.3. Interobserver and intraobserver evaluations were calculated using Cohen's kappa and Fleiss' kappa, respectively, using R V.4.0.3 (2020-10-10) (packages 'irr').

**Table 1** SCORDoK terminology[14]

| SCORDoK skin change | Description |
| --- | --- |
| Telogen effluvium | Loss of hair<br>Yes (if present) and no (if absent) |
| Pigmentary changes | Any and all pigmentary changes on the skin observed (hypopigmentation and hyperpigmentation)<br>Yes (if present) and no (if absent) |
| Ichthyosiform skin changes | Grade 1: Dry hyperpigmented skin with prominent skin lines and greyish scale<br>Grade 2: Hyperkeratotic areas and thick greyish scales<br>Grade 3: Hard to shiny hyperpigmented thick scaling which leaves erosions on detachment |
| Lichenoid skin changes | Grade 1: 1–5 mm hyperpigmented to purple-brown flat papules<br>Grade 2: variable-sized hyperpigmented, hyperkeratotic and lichenified well-defined plaques<br>Grade 3: thicker infiltrated plaques which may be shiny. May detach, leaving erosions or intact epidermis beneath |
| Bullae, erosions and desquamations | Grade 1: presence of bullae, erosions and desquamation affecting <5% body surface area<br>Grade 2: presence of bullae, erosions and desquamation affecting 5%–30% body surface area<br>Grade 3: presence of bullae, erosions and desquamation affecting >30% body surface area |

## RESULTS

### Descriptive statistics

Study flow is shown in figure 1. Of children enrolled in CHAIN but not included in the skin study (101 children), 4 had KW (4%), 19 SW (19%), 12 MW (12%) and 66 NW (65%). From the CHAIN dataset, 94/101 had no visible skin changes while 7 children presented with dermatoses: depigmentation (2), rash with broken skin (2), dermatitis (2) and maculopapular skin rash (1).

In total 103 children were included in the skin study with a median age of 12 months (table 2). 11 (11%) children had KW, 31 children (30%) had SW, 20 (19%) had MW while 41 (40%) had NW. 48 (47%) children were admitted with diarrhoea, and 18 (18%) had a positive HIV antibody test. Dermatologists (using SCORDoK) determined 51 of the 103 (%) children had a dermatoses.

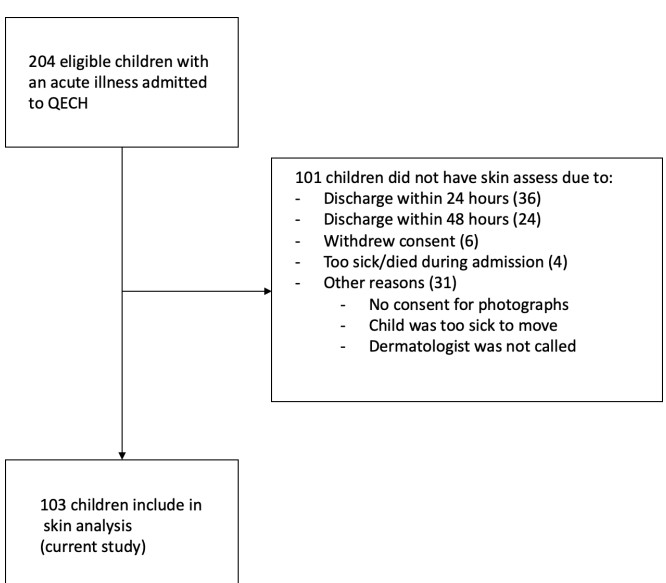

**Figure 1** Study recruitment flow diagram. QECH, Queen Elizabeth Central Hospital.

### Skin changes according to SCORDoK

Pigmentary changes were the most frequently observed classification (34%) using SCORDoK, with telogen effluvium the second most prevalent (table 2). Ichthyosiform skin changes were seen in two children with NW and lichenoid skin changes in one child with nutritional oedema (kwashiorkor) (5%). Skin changes were more prevalent in children with nutritional oedema (kwashiorkor) except ichthyosiform skin changes. Children with SW were more likely to present with telogen effluvium or bullae, erosions and desquamation compared with pigmentary changes. 52 children (50%) were graded as having no skin changes. Of the six children who died, three (50%) children presented with pigmentary changes, two (33%) with telogen effluvium and one with bullae, erosions and desquamation.

Skin changes of children with severe malnutrition were compared with skin changes reported by Heilskov et al in hospitalised Ugandan children (table 3).[14] Pigmentary changes were as prevalent in both studies among children with nutritional oedema (kwashiorkor) but children included in our analysis did not have the same prevalence of ichthyosiform and lichenoid skin changes. Dermatologists found a significantly larger percentage of patients with nutritional oedema (kwashiorkor) and SW had bullae, erosions and desquamation compared with Heilskov et al.[14] Rates of telogen effluvium were significantly lower in children with nutritional oedema (kwashiorkor) and SW in our analysis.

### Dermatological diagnoses

The leading diagnoses made were congenital dermal melanocytosis, diaper dermatitis and eczema, each in at least 10% of children (table 4). There were three (27%) cases each of congenital dermal melanocytosis and flaky paint dermatoses in children with nutritional oedema (kwashiorkor). Children with SW presented most

**Table 2** Baseline characteristics of 103 children enrolled in skin CHAIN study and recruited for the current study

|  | All | KW | SW | MW | NW | P value |
|---|---|---|---|---|---|---|
| N (%) | 103 | 11 (10.7) | 31 (30.1) | 20 (19.4) | 41 (39.8) |  |
| Mean age in months (SD) | 12.4 (±5.9) | 19.1 (±4.1) | 11.1 (±6.1) | 10.5 (±5.1) | 11.9 (±5.8) | <0.001* |
| WLZ (SD) | −1.4 (±1.8) | −1.6 (±1.3) | −3.2 (±1.1) | −1.5 (±1.3) | 0.1 (±1.1) | <0.001* |
| MUAC (SD) | 12.5 (±1.9) | 12.3 (±1.2) | 10.5 (±1.1) | 12.3 (±0.6) | 14.3 (±1.1) | <0.001* |
| HIV antibody positive (%) | 18 (17.5) | 1 (9.1) | 8 (25.8) | 7 (35.0) | 2 (4.9) | 0.01† |
| Sex=male (%) | 63 (61.2) | 5 (45.5) | 20 (64.5) | 14 (70.0) | 24 (58.5) | 0.56† |
| Mortality (%) | 6 (5.8) | 0 | 5 (16.1) | 1 (5.0) | 0 | 0.03† |
| Diarrhoea (%) | 48 (46.6) | 6 (54.5) | 17 (54.8) | 11 (55.0) | 14 (34.1) | 0.24† |
| Skin changes according to SCORDoK: |  |  |  |  |  |  |
| TE‡ (%) | 20 (19.4) | 5 (45.5) | 9 (29.0) | 5 (25.0) | 1 (2.4) | 0.002† |
| PC (%) | 35 (34.0) | 8 (72.7) | 7 (22.6) | 7 (35.0) | 13 (31.7) | 0.03† |
| ISC (%) | 2 (1.9) | 0 | 0 | 0 | 2 (4.9) | 0.38† |
| LSC (%) | 1 (1.0) | 1 (9.1) | 0 | 0 | 0 | 0.04† |
| BED (%) | 18 (17.5) | 6 (54.5) | 9 (29.0) | 1 (5.0) | 2 (4.9) | <0.001† |

See table 1 for explanation on the dermatological descriptions of SCORDoK.
*One way ANOVA.
†χ².
‡TE was graded as hair loss.
ANOVA, analysis of variance; BED, bullae, erosions and desquamation; CHAIN, Childhood Acute Illness and Nutrition; ISC, ichthyosiform skin changes; KW, kwashiorkor; LSC, lichenoid skin changes; MUAC, mid-upper arm circumference; MW, moderate wasting; NW, no wasting; PC, pigmentary changes; SW, severe wasting; TE, telogen effluvium; WLZ, weight for length Z-score.

frequently with diaper dermatitis (22.6%), eczema (19.4%) and congenital dermal melanocytosis (19.4%) In children with MW, congenital dermal melanocytosis followed by popular urticaria were the most common dermatoses seen. The most common diagnoses in children with no NW were congenital dermal melanocytosis, diaper dermatitis and postinflammatory hyperpigmentation.

The majority of postinflammatory hyperpigmentation was considered due to insect bites (papular urticaria). The flag sign was not seen, but thinning and discolouration of hair were seen in three children (2.9%); one with nutritional oedema (kwashiorkor), one with SW and one with MW. The most common dermatological complaint reported prior to admission was a rash, seen in 29 children (28%), followed by complaints of itching (15 cases, 15%), swelling (9 cases, 8.7%) and ulcers (7 cases, 6.8%). In children with SW, 45% reported dermatological complaints prior to admission: these included rash (32%), itching (9.7%), ulcers (9.7%) and swelling (13%). In children with nutritional oedema (kwashiorkor), four reported a rash (36%), and one (9.1%) reported complaints of itching prior to admission. 40% of children (41) did not have a clinical dermatological diagnosis.

**Follow-up**

Of the 103 children, 101 children were discharged alive. There were an additional four deaths postdischarge,

**Table 3** Frequency of skin disorders in 42 severely malnourished children admitted to hospital in Blantyre, Malawi compared with 119 admitted in Kampala, Uganda

|  | Malawi KW (11) van den Brink et al (current study) | Uganda KW (77) Heilskov et al[14] | P value | Malawi SW (31) van den Brink et al (current study) | Uganda SW (42) Heilskov et al[14] | P value |
|---|---|---|---|---|---|---|
| TE* | 45% | 83% | 0.01 | 29% | 69% | <0.001 |
| PC | 73% | 73% | 0.99 | 23% | 40% | 0.13 |
| ISC | 0 | 30% | 0.06 | 0 | 5% | 0.50 |
| LSC | 9% | 25% | 0.68 | 0 | 5% | 0.50 |
| BED | 55% | 24% | 0.03 | 29% | 0% | <0.001 |

*TE was graded as hair loss.
BED, bullae, erosions and desquamation; ISC, ichthyosiform skin changes; KW, kwashiorkor; LSC, lichenoid skin changes; PC, pigmentary changes; SW, severe wasting; TE, telogen effluvium.

van den Brink D, et al. BMJ Paediatrics Open 2024;8:e002289. doi:10.1136/bmjpo-2023-002289

**Table 4** Diagnoses made by dermatologists from photographs

| Diagnosis | # of children | % | KW | SW | MW | NW |
|---|---|---|---|---|---|---|
| Congenital dermal melanocytosis | 19 | 18.4% | 3 (27.3%) | 6 (19.4%) | 5 (25%) | 5 (12.2%) |
| Diaper dermatitis | 16 | 15.5% | 2 (18.2%) | 7 (22.6%) | 2 (10%) | 5 (12.2%) |
| Eczema | 11 | 10.7% | 1 (9.1%) | 6 (19.4%) | 1 (5%) | 3 (7.3%) |
| Postinflammatory hyperpigmentation | 10 | 9.7% | 1 (9.1%) | 3 (9.7%) | 1 (5%) | 5 (12.2%) |
| Papular urticaria | 7 | 6.8% | 1 (9.1%) | 2 (6.5%) | 3 (15%) | 1 (2.4%) |
| Postinflammatory hypopigmentation | 4 | 3.9% | 1 (9.1%) | 0 | 2 (10%) | 1 (2.4%) |
| Tinea capitis | 4 | 3.9% | 1 (9.1%) | 1 (3.2%) | 2 (10%) | 0 |
| Oral thrush | 3 | 2.9% | 2 (18.2%) | 1 (3.2%) | 0 | 0 |
| Angular cheilitis | 3 | 2.9% | 1 (9.1%) | 1 (3.2%) | 0 | 1 (2.4%) |
| Flaky paint dermatoses | 3 | 2.9% | 3 (27.3%) | 0 | 0 | 0 |
| Congenital melanocytic nevi | 1 | 1.0% | 0 | 1 (3.2%) | 0 | 0 |
| Papular prurigo | 1 | 1.0% | 0 | 0 | 1 (5%) | 0 |
| Skin ulcer | 1 | 1.0% | 0 | 1 (3.2%) | 0 | 0 |
| Scabies | 1 | 1.0% | 0 | 0 | 0 | 1 (2.4%) |
| Café au lait spot | 1 | 1.0% | 0 | 0 | 0 | 1 (2.4%) |
| Milia | 1 | 1.0% | 0 | 0 | 1 (5%) | 0 |
| Naevus | 1 | 1.0% | 0 | 0 | 0 | 1 (2.4%) |
| Acrodermatitis enterohepatica | 1 | 1.0% | 0 | 1 (3.2%) | 0 | 0 |

Percentages are per nutritional strata; 103 children included in total, KW 11 children, SW 31 children, MW 20 children, NW 41 children.
KW, kwashiorkor; MW, moderate wasting; NW, no wasting; SW, severe wasting.

before the 180-day follow-up postdischarge. A total of 60 children were assessed again at 180 days after discharge, 15 lost to follow-up, 22 unable to have photos taken and 6 children who died since enrolment. At 180 days, there was a lower prevalence of telogen effluvium compared with admission (table 5). Bullae, erosions and desquamation had a reduction in prevalence in children with nutritional oedema (kwashiorkor). Paired analysis on follow-up showed that 24 (40%) children had no change in their skin status from admission to follow-up. Six (12%) of children presented with novel cases of telogen effluvium and 11 (18%) with new pigmentary changes. There was one new case of ichthyosiform and lichenoid skin changes (each) at follow-up. Across all four nutritional strata, there was an improvement in skin changes that were seen at admission compared with follow-up.

### Interobserver and intraobserver variability

103 assessments at admission and 60 at day 180 after discharge were included in the inter-rater and intrarater variability analysis. There was a moderate to fair Fleiss' kappa for inter-rater variability between the SCORDoK scores of the three dermatologists (table 6). Bullae, erosions and desquamation had a moderate agreement by Fleiss' kappa while others had slight to fair agreement.[22] Intrarater variability among dermatologists using Cohen's kappa was one fair with agreement, and two

**Table 5** Prevalence of skin changes of 60 patients on admission compared with follow-up at 180 days postdischarge using SCORDoK

| | Admission (60 patients) | Day 180 (60 patients) | Admission Kwash (8) | Admission SW (12) | Day 180 Kwash (8) | Day 180 SW (12) |
|---|---|---|---|---|---|---|
| TE* | 20% | 12% | 50% | 25% | 12.5% | 25% |
| PC | 32% | 32% | 50% | 25% | 25% | 16.7% |
| ISC | 0% | 1.7% | 0% | 0% | 0% | 0% |
| LSC | 0% | 1.7% | 0% | 0% | 12.5% | 0% |
| BED | 13% | 5.0% | 25% | 16.7% | 0% | 0% |

*TE was graded as hair loss.
BED, bullae, erosions and desquamation; ISC, ichthyosiform skin changes; LSC, lichenoid skin changes; PC, pigmentary changes; SW, severe wasting; TE, telogen effluvium.

raters with moderate agreement when grading pictures for a second time.[22]

## DISCUSSION

In this Malawian population of hospitalised young children, stratified according to nutritional status, the prevalence of skin changes in the subgroup of malnourished children was unexpectedly lower than reported in Uganda.[14] In general, skin changes in the present group of acutely ill children, stratified according to nutritional status were not very prevalent. The hospitals in Kampala (where SCORDOK was developed) and Blantyre are similar in many ways,[23] both studies prospectively recruited severely malnourished children using similar WHO criteria for severe malnutrition. Our study focused on children who were admitted to the hospital with an acute illness rather than due to severe malnutrition alone, and we excluded early deaths and children unable to have photography. These biases could partly explain the lower prevalence of dermatoses seen in this study. Prior published studies may also have suffered from observation and selection bias, resulting in a higher reported frequency of skin abnormalities in children with severe acute malnutrition (SAM).[6 14 24 25] Improvements in the prevention of malnutrition with initiatives such as community nutrition programmes,[26–28] sanitation and hygiene interventions,[26–30] and investments in agriculture and reducing economic inequity,[26 27] could also have contributed to a lower prevalence of dermatoses in our setting.

Pigmentary changes, telogen effluvium and bullae erosive desquamation were most often observed in children with nutritional oedema (kwashiorkor). We had expected to see higher rates of pigmentary changes as nutritional oedema (kwashiorkor) can alter pigmentation through thinning of the skin resulting in hypopigmentation, as well as delayed wound healing causing hyperpigmentation.[31] We also expected to see more lichenoid skin changes as they have previously been associated with nutritional oedema (kwashiorkor).[14 32] This association has been suggested to be a skin manifestation of niacin deficiency[33] due to the decreased levels tryptophan as a direct result of increased levels of proinflammatory cytokine IFN-γ.[33 34] Pigmentary changes were not specific to children with SW or nutritional oedema (kwashiorkor) but were also present in

children with MW or NW. Telogen effluvium, or hair thinning, as well as bullae, erosions and desquamation were less prevalent 180 days after discharge, suggestive of an association with acute illness and/or malnutrition. The most common diagnoses seen, congenital dermal melanocytosis, diaper dermatitis, eczema and postinflammatory hyperpigmentation are not specific to malnutrition and the reported prevalence is similar to the global prevalence of dermatoses in infants.[35–38]

Observed differences in the interpretation of the SCORDoK tool were reflected by poor kappa scores with ichthyosiform skin change, telogen effluvium and lichenoid skin change having the lowest agreement, which were subsequently categories stimulating the most discussions among the dermatologists. Bullae, erosions and desquamation had the highest Fleiss kappa score but with only moderate agreement. There was consensus that the category telogen effluvium did not allow for proper classification of what should be thinning and discolouration of hair, which is specific to malnutrition. Telogen effluvium, interpreted as loss of hair/alopecia, was most commonly along the occiput parietal temporal area. This is where the cloth used to carry babies on mothers' backs would cause friction and is the likely cause of the alopecia. In addition to friction, several reasons for hair loss in childhood exist that are not necessarily due to malnutrition (ie, cultural, mechanical and fungal infections).[39] Using telogen effluvium as a marker of malnutrition is, therefore, likely to be unreliable. Pigmentary changes were a broad category that resulted in congenital pigmentary dermatoses, such as congenital dermal melanocytosis, being classified as a pigmentary change. Both ichthyosiform skin changes and lichenoid skin changes were broad categories that lacked specific criteria. The SCORDOK dermatological tool was developed for children with SAM between 6 and 59 months, and the children in this study were below 24 months. We believe that the two populations are similar enough since the majority of children with SAM are below 2 years of age.

Challenges with dermatological grading tools are not unique, as skin changes can be subject to different interpretations and various classifications.[40] SCORAD, a skin grading tool developed to determine the severity of atopic dermatitis,[40] was found to have similar intraobserver and interobserver agreements. Subjectivity in

**Table 6** (A) Inter-rater variability with Fleiss' kappa; (B) intrarater variability with Cohen's kappa

| | **(A) Inter-rater variability** | | | | | | **(B) Intrarater variability** | | |
| --- | --- | --- | --- | --- | --- | --- | --- | --- | --- |
| | **TE*** | **PC** | **ISC** | **LSC** | **BED** | **Total** | **Grader 1** | **Grader 2** | **Grader 3** |
| Kappa | 0.18 | 0.41 | 0.05 | 0.20 | 0.46 | 0.25 | 0.55 | 0.58 | 0.47 |
| P value | 0.001 | <0.001 | 0.42 | 0.21 | <0.001 | | <0.001 | <0.001 | <0.001 |

*TE was graded as hair loss.
BED, bullae, erosions and desquamation; ISC, ichthyosiform skin changes; LSC, lichenoid skin changes; PC, pigmentary changes; TE, telogen effluvium.

van den Brink D, *et al. BMJ Paediatrics Open* 2024;**8**:e002289. doi:10.1136/bmjpo-2023-002289

SCORAD ratings potentially arose from social and cultural factors.[40]

This prospective study reported on observed dermatoses in a population of acutely ill children across a range of nutritional status admitted to a hospital in Malawi. The dermatologists involved with the grading of dermatoses have extensive dermatological experience in treating dark skin. This study suffered from selection bias due to the photographic requirements that required additional consent and that children were medically stable (figure 1). At admission to the main CHAIN study, all children were assessed by a clinician including a skin assessment. This bias is limited as only seven children who were not included were reported to have skin dermatoses. The sample size for severe malnutrition was limited with only 41% of children included having SW or nutritional oedema (kwashiorkor). Despite standardised photography, there were challenges in classifying skin disorders in this way. Our dermatologists highlighted the importance of patient histories that also affect diagnoses of dermatoses. Lastly, the interobserver assessment was completed 6 months after initial assessment and intervening discussion may have influenced the second grading set potentially resulting in a higher interobserver variability.

In conclusion, skin changes in hospitalised children with an acute illness and stratified according to nutritional status were not as prevalent as historically reported.[6 8 14 24 25] The high disagreement rates in scoring also highlight difficulties with interpretive diagnostic tools. A future alternative for grading dermatoses could be machine learning, in order to more objectively classify skin changes.

## Author affiliations
[1]Amsterdam Centre for Global Child Health & Emma Children's Hospital, Pediatrics, Amsterdam UMC, Locatie AMC, Amsterdam, The Netherlands
[2]Department of Dermatology, Queen Elizabeth Central Hospital, Blantyre, Southern Region, Malawi
[3]Caribbean Institute for Health Research, University of the West Indies, Kingston, Jamaica
[4]Department of Dermatology, Erasmus Medical Center, Rotterdam, The Netherlands
[5]Department of Paediatrics and Child Health, Kamuzu University of Health Sciences, Blantyre, Southern Region, Malawi
[6]Division of Gastroenterology, Hepatology and Nutrition, The Hospital for Sick Children, Toronto, Ontario, Canada
[7]Childhood Acute Illness Network, Nairobi, Kenya
[8]Departments of Global Health, Epidemiology, Infectious Disease, University of Washington, Seattle, Washington, USA
[9]Department of Global Health, Amsterdam Institute for Global Health and Development, Amsterdam, Netherlands
[10]KEMRI-Wellcome Trust Research Programme, Kilifi, Kenya
[11]Centre for Tropcial Medicine & Global Health, Oxford University, Oxford, UK

**Acknowledgements** We thank all the children and their families for taking part in the study, and for all the commitments from the staff at QEH. Lastly, we would like to thank FOX-IT for providing access to their servers. For the purpose of open access, the CHAIN Network has applied a CC BY public copyright licence to any author-accepted manuscript version arising from this submission.

**Contributors** Conceptualisation: WV, KM, CvH and JB. Data acquisition: DvdB, KM, CvH, DT, FN, ES and EM. Analysis and interpretation of the data: DvdB, KM, CvH, DT, MBvH, RHJB, JLW, DB, JB and WV. Original manuscript draft: DvdB and WV. Review and editing: DvdB, KM, CvH, DT, FN, ES, EM, MBvH, RHJB, JLW, DB, JB and WV. All authors approved the final version of the manuscript and are accountable for all aspects of the work. Guarantors for the overall content: DvdB.

**Funding** Support for the CHAIN Network was provided by the Bill & Melinda Gates Foundation (OPP1131320).

**Competing interests** None declared.

**Patient and public involvement** Patients and/or the public were not involved in the design, or conduct, or reporting, or dissemination plans of this research.

**Patient consent for publication** Not applicable.

**Ethics approval** This study involves human participants and ethical approval was obtained from Oxford University Tropical Research Ethics Committee, COMREC, Kamuzu University of Health Sciences, Malawi, and Medical Ethics Review Committee, Amsterdam UMC, The Netherlands. Participants were minors, informed consent was obtained from their parents or caregivers.

**Provenance and peer review** Not commissioned; internally peer reviewed.

**Data availability statement** Data are available at a reasonable request from https://dataverse.harvard.edu/dataverse/chain

**ORCID iDs**
Deborah van den Brink http://orcid.org/0000-0002-0458-0669
Debbie Thompson http://orcid.org/0000-0001-6781-7646
Judd L Walson http://orcid.org/0000-0003-4836-720X

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
