## [Reviewer comments · BMJ Paediatrics Open]

ARTICLE DETAILS

TITLE (PROVISIONAL)	Dermatological changes in a prospective cohort of acutely ill, hospitalized Malawian children, stratified according to nutritional status
AUTHORS	van den Brink, Deborah Mponda , Kelvin Thompson, Debbie van Hees, Colette Ngong'a, Fletcher Segula, Emma Mbale, Emmie Boele van Hensbroek , Michael Bandsma, Robert H. J. Walson, Judd L. Brals, Daniella berkely , james Voskuil, Wieger

VERSION 1 – REVIEW

REVIEWER	Dr. Mark J. Manary Washington Univ United Kingdom of Great Britain and Northern Ireland
REVIEW RETURNED	02-Dec-2023

GENERAL COMMENTS	This is a useful addition to the malnutrition literatue. I am surprised that you are not including color photos, they would be SO useful. Please consider this.
---

REVIEWER	Prof. Piyush Gupta 33A, Royal Apartments Sector Sigma IV Greater Noida, UP Greater Noida Uttar Pradesh 201310 India
REVIEW RETURNED	14-Mar-2024

GENERAL COMMENTS	The study title implies a descriptive analysis of skin changes in children with severe malnutrition. The authors fail to achieve the primary objective due to following observations: 1. The term “severe malnutrition” is not defined in Methods. Authors have defined “severe wasting” and “moderate wasting”. WHO has defined severe malnutrition as those having severe wasting or those having unexplained bilateral pitting oedema. The results (shown in Table 3) imply that there were only 42 severely malnourished children in the present study.
--

	2. The sample size finally achieved in this study (n=42) (including 31 as severe wasting and 11 labelled as kwashiorkor) is too small to reflect and portray the entire spectrum of skin changes that occur in these children. It is also well known that children with Kwashiorkor have the highest prevalence of changes. In the present study, there were only 11 children with oedema. It is important to note that these children were enrolled as having an acute illness; thus, they could be having oedema due to other causes as well, and may not necessarily represent Kwashiorkor. 3. A major concern while including patients with severe malnutrition for dermatological assessment is the convenience sampling due to which almost 50% of the eligible children (101/204) were NOT assessed for skin condition because of various logistics reasons. This greatly undermines the validity of the study results. 4. Another objective was to validate the SCORDOK grading tool in Malawian setting. It is to be noted that this score was developed for children with severe acute malnutrition between the ages of 6 mo-5years. While in the present study, it has been applied on children aged 1 month-23 months. The aetiology and clinical manifestation of SAM below 6 months is entirely different from older infants and children. The original score for children > 6 months age has not been validated in younger age group, so its application in the current setting is questionable. 5. I was confused as to why the results are presented for all children with acute illness (including even those which were not malnourished).
--	--

VERSION 1 – AUTHOR RESPONSE

Dear Bolajoko Olusanya and Shanti Raman,

We are grateful for the opportunity to re-submit our manuscript to BMJ Paediatrics Open, after the recent review.

We have replied to all issues raised by the reviewers (see attached document) and have put our replies to their concerns in the attached document.

We think it is important to emphasize that the CHAIN network and its study activities were never a '(severe-) malnutrition study' but rather a study with the aim to explore pathways underlying vulnerability and increased risk on poor outcome in young (<2 years) children admitted to a hospital with an acute illness in a low resource setting. Malawi was one of 9 sites in 6 countries in both Africa and South-East Asia. We enrolled prospectively and therefore the sample indeed contains acutely ill, young children including children with SAM since these children have a high risk of illness. We have adjusted/reframed the message of our manuscript accordingly and this can be found on page 3 as well as the discussion of the revised manuscript. We believe this will sufficiently address this point raised by both you and one of the reviewers.

We hope that we have successfully done so and look forward to hearing from you.

Sincerely,

Deborah van den Brink and Wieger Voskuil (on behalf of the Skin CHAIN team)

VERSION 2 – REVIEW

VERSION 2 – AUTHOR RESPONSE

Dear Bolajoko Olusanya and Shanti Raman,

Thank you so much for your review and comments of the latest version of our manuscript. We have reviewed the paper for language and grammar, and have removed the italics from the abstract and key findings section. We rewrote the 'Skin assessment through photography' statement, and have presented our key findings in point form.

We have updated the response letter to the reviewer and editors with these comments added at the end.

Sincerely,

Deborah van den Brink and Wieger Voskuil (on behalf of the Skin CHAIN team)

VERSION 3 – REVIEW

REVIEWER	Dr. Mark J. Manary Washington Univ United Kingdom of Great Britain and Northern Ireland
REVIEW RETURNED	13-May-2024

GENERAL COMMENTS	It is now much clearer who was included in the study population. Skin findings always stand out to clinician. Many clinicians have firm opinions on the interpretations of the significance of a particular finding , This study is humbling in this regard, showing that many of these findings are seen in diversity of settings.
---

REVIEWER	Prof. Piyush Gupta 33A, Royal Apartments Sector Sigma IV Greater Noida, UP Greater Noida Uttar Pradesh 201310 India
REVIEW RETURNED	13-May-2024

GENERAL COMMENTS	none
------

VERSION 3 – AUTHOR RESPONSE

None